# Peptide Engraftment on PEGylated Nanoliposomes for Bone Specific Delivery of PTH (1-34) in Osteoporosis

**DOI:** 10.3390/pharmaceutics15020608

**Published:** 2023-02-11

**Authors:** Sagar Salave, Suchita Dattatray Shinde, Dhwani Rana, Bichismita Sahu, Hemant Kumar, Rikin Patel, Derajram Benival, Nagavendra Kommineni

**Affiliations:** 1National Institute of Pharmaceutical Education and Research (NIPER), Ahmedabad 382355, India; 2Intas Pharmaceuticals Ltd., Matoda Village, Ahmedabad 382210, India; 3Center for Biomedical Research, Population Council, New York, NY 10065, USA

**Keywords:** osteoporosis, PTH (1-34), anabolic peptide, Targeting, bone, Central Composite Design

## Abstract

Bone-specific functionalization strategies on liposomes are promising approaches to delivering the drug in osteoporotic conditions. This approach delivers the drug to the bone surface specifically, reduces the dose and off-target effects of the drug, and thereby reduces the toxicity of the drug. The purpose of the current research work was to fabricate the bone-specific peptide conjugated pegylated nanoliposomes to deliver anabolic drug and its physicochemical evaluations. For this, a bone-specific peptide (SDSSD) was synthesized, and the synthesized peptide was conjugated with a linker (DSPE-PEG2000-COOH) to obtain a bone-specific conjugate (SDSSD-DSPE). Purified SDSSD-DSPE was characterized by HPLC, Maldi-TOF, NMR, and Scanning Electron Microscope/Energy Dispersive Spectroscopy (SEM/EDS). Further, peptide-conjugated and anabolic drug-encapsulated liposomes (SDSSD-LPs) were developed using the ethanol injection method and optimized by Central Composite Design (CCD) using a statistical approach. Optimized SDSSD-LPs were evaluated for their physicochemical properties, including surface morphology, particle size, zeta potential, in vitro drug release, and bone mineral binding potential. The obtained results from these studies demonstrated that SDSSD-DSPE conjugate and SDSSD-LPs were optimized successfully. The particle size, % EE, and zeta potential of SDSSD-LPs were observed to be 183.07 ± 0.85 nm, 66.72 ± 4.22%, and −25.03 ± 0.21 mV, respectively. SDSSD-LPs demonstrated a sustained drug release profile. Further, the in vitro bone mineral binding assay demonstrated that SDSSD-LPs deliver the drug to the bone surface specifically. These results suggested that SDSSD-LPs could be a potential targeting approach to deliver the anabolic drug in osteoporotic conditions.

## 1. Introduction

Good bone health essentially resides in maintaining adequate levels of bone mass. Bone density is gradually lost upon aging, soon after reaching the peak bone mass between the ages of 25 and 30. Upon this consideration of bone mineral density (BMD), the World Health Organization (WHO), based on T-score BMD, states that osteoporosis exists when BMD lies 2.5 standard deviations (SD) or more below the average value for healthy young women (T-score of <−2.5 SD) [1]. Osteoporosis, a musculoskeletal disorder, constitutes an enormous health burden on society [2]. The loss of BMD roots leads to the progression of microarchitectural deterioration and leads to the likelihood of fragility fractures. This malady often advances silently without the manifestation of noticeable symptoms until the incidence of fracture occurs. Accelerated osteoclastic bone resorption plays a principal part in the pathogenesis of osteoporosis. For the management of disease, several effective pharmacological interventions are available; antiresorptive and osteoanabolic. Antiresorptive agents, viz. bisphosphonates, estrogen, RANK ligand inhibitors, selective estrogen receptor modulators, calcitonin, and monoclonal antibodies cause suppression of osteoclastic-mediated bone resorption and decrease in bone turnover, whereas anabolic agents, namely teriparatide, abaloparatide, and Romosumab, assist in forming new bone by activating osteoblasts and bone remodeling [3]. For the past several decades, antiresorptive therapy has remained the mainstay of therapy for osteoporosis. However, improvement in the persisting bone quality results when, apart from a reduction in the rate of resorption, the formation of new bone is also enhanced. Osteoanabolics attribute this therapeutic action primarily to stimulating the osteoblastic formation of fresh bone on the quiescent bone lining that is not concurrently going through bone remodeling [4]. Anabolic agents are capable of restoring the BMD, filling in the bone remodeling space, actively raising the bone mass, and decreasing the consequences of osteoporotic fractures [5]. Moreover, resorption inhibitors such as alendronate and other bisphosphonates have been linked to a greater prevalence of depressive symptoms in recent case reports [6].

PTH (1-34) (Teriparatide) is a recombinant/synthetic analog of parathyroid hormone (PTH) and acts by binding to PTH receptors with similar affinity while displaying similar physiological actions. This property is imparted owing to the molecular structure of PTH (1-34), the first 34 amino acids of the intact PTH [7]. Intermittent exposure to PTH (1-34) is advisable for the formation of bone through stimulating osteoblasts predominantly on trabecular and cortical bone surfaces [8]. PTH (1-34) is recommended for treating osteoporosis in postmenopausal women at high risk of fracture as well as for treating osteoporosis in both men and women who have had persistent systemic glucocorticoid therapy. When administered as a daily subcutaneous injection, this 20-mcg dose is not recommended for more than two years during the course of the patient’s life [7]. However, frequent and repeated injections compromise patient compliance. All these factors demand a new targeted delivery system for PTH (1-34) for better results. Therefore, bone-targeted delivery of PTH (1-34) can be accomplished by nanovesicles conjugated with bone-targeted peptides, which can deliver the drug to the bone without producing side effects [9,10]. Several reports suggest the role of amino acids in bone mineralization [11,12,13]. Non-collagenous proteins express negatively charged amino acids like glutamic acid and aspartic acid, which are essential for controlling the nucleation and development of hydroxyapatite (HA) [14]. For instance, (Asp-Ser-Ser)_6_ showed favorable binding to low crystallized HA, which is known as the bone formation zone [15,16], whereas bone resorption (highly crystalized HA) specific targeting has been achieved by (Asp)_8_.

SDSSD (Ser-Asp-Ser-Ser-Asp) peptide showed tremendous potential in bone formation as well as bone targeting. For instance, the SDSSD peptide was immobilized on the zirconia implant surface to enhance osteoblast bioactivity [17]. Cai Mingxiang et al. reported that SDSSD-modified 3D bio scaffolds facilitate osteogenesis and bone formation in the subcutaneous pocket of BALB/c nude mice and facilitate bone healing in vivo [18]. The author has also reported that bone formation was promoted by binding SDSSD to the G protein-coupled receptor and regulating the AKT signaling pathway. In another study, an SDSSD-modified chitosan scaffold promoted the intramembranous ossification bone repair process [19]. For bone-specific targeting, Yao Sun et al. have developed SDSSD-modified polyurethane nano micelles (SDSSD-PU) encapsulated with siRNA/microRNA to treat osteoblast-induced bone diseases [20]. The author has reported that SDSSD-PU could be used to target the therapeutic agent not only on the bone formation surface but also at the osteoblast. In another study by Cui Yongzhi et al., exosomes modified with SDSSD were developed for bone-targeted delivery of siRNA to treat osteoporosis [21]. Liu Meijing et al. prepared SDSSD-modified geniposidic acid conjugate to target the bone and promote osteogenesis [22]. Additionally, SDSSD-modified ferritin nanoparticles have also been explored for bone-targeted imaging [23]. Based on this knowledge, it can be concluded that the SDSSD peptide has not only bone-targeting capabilities but also helps in osteogenesis.

Several nanoformulations showed bone regeneration as well as osteogenic differentiation potential [24,25]. Liposomes are lipid-based vesicular systems characterized by a lipidic bilayer and an internal aqueous cavity. These types of formulations have revolutionized the drug delivery field and have successfully translated into real-time clinical applications [26,27,28,29]. Therefore, the present work aims to develop SDSSD-conjugated bone-targeted pegylated nanoliposomes (SDSSD-LPs) for PTH (1-34) delivery in osteoporosis. A novel drug delivery approach could deliver the drug specifically to the bone and minimize the side effects compared to unconjugated nanocarriers.

## 2. Materials and Methods

PTH (1-34), cholesterol, DSPE-PEG2000-COOH (1,2-distearoyl-sn-glycero-3-phosphoethanolamine-N-[carboxy(polyethylene glycol)-2000] (sodium salt), N-Hydroxysuccinimide (NHS), N-(3-Dimethylaminopropyl)-N′-ethyl carbodiimide hydrochloride (EDC.HCl) and hydroxyapatite were purchased from Sigma-Aldrich (Bangalore, India). N-(carbonyl methoxy polyethyleneglycol-2000)-1,2-distearoylsn-glycero-3-phosphoethanolamine (Na-salt; MPEG-2000-DSPE) [DSPE-PEG2000] and hydrogenated phosphatidylcholine (HSPC) were received as gift samples from Lipoid GmbH (Ludwigshafen am Rhein, Germany). Sodium hydrogen phosphate, potassium chloride, potassium dihydrogen phosphate, isopropyl alcohol, formic acid, and acetonitrile were purchased from Fischer Scientific (Mumbai, India). Absolute ethanol was obtained from Shree Chalthan Vibhag Khand, Uddyog Sahakari Mandli Ltd., Surat, Gujarat. NBD-PE (N-(7-Nitrobenz-2-Oxa-1,3-Diazol-4-yl)-1,2-Dihexadecanoyl-sn-Glycero-3 Phosphoethanolamine, Triethylammonium Salt) was procured from InvitrogenTM, Thermo-Fischer Scientific (Mumbai, India). HiMedia Laboratories Pvt. Ltd. provided sodium chloride (Mumbai, India). The use of all other reagents was done without further processing, and all of them were of analytical grade.

### 2.1. Synthesis of SDSSD

The pentapeptide (SDSSD) was synthesized via the standard protocol of solid-phase peptide synthesis on an automated peptide synthesizer (BiotageAlstra). Amino acid serine and aspartic acids were orthogonally protected and coupled on the solid support of 2-chlorotrityl resins by employing the use of standard coupling reagents for Fmoc chemistry. Briefly, 2-chlorotrityl resins were swelled in dichloromethane prior to the coupling of amino acids. Fmoc-Asp(OtBu)-OH was coupled with resins in the presence of a base, diisopropylethylamine (DIPEA). Next to coupling, deprotection of the Fmoc group was carried out using 20 % *v*/*v* piperidine solution in dimethylformamide (DMF). Coupling of the second amino acid (Fmoc-Ser-OH) was done in the presence of coupling reagent ([Bis (dimethylamino)methylene]-1H-1,2,3-triazolo [4,5-b]pyridinium3-oxid hexafluorophosphate) and DIPEA as a base in DMF. Deprotection of the Fmoc group and coupling cycle was repeated until the last amino acid sequence. The peptide was cleaved using 90% *v*/*v* trifluoroacetic acid in deionized water along with 2% *v*/*v* triisopropylsilane as cationic scavenger, and precipitation was done in cold diethyl ether. The purification of the peptide was done on Agilent semi-preparative Reverse Phase-HPLC over X bridge BEH-C18 peptide column (300 Å, 4.6 mm × 150 mm, 10 μm,) followed by characterization by LCMS and ^1^H NMR [30,31]. The purified peptide was lyophilized and stored at −20 °C until further use.

### 2.2. Synthesis of Conjugate (SDSSD-DSPE)

A lipid-PEG-peptide combination that may subsequently be added to liposomes for bone targeting was developed. SDSSD was conjugated to DSPE-PEG2000-COOH using carbodiimide chemistry, based on the previously described method by Wang et al. with modifications [32]. Briefly, DSPE-PEG2000-COOH was dissolved in 0.1 M MES buffer at pH 6.0 and stirred at 2–8 °C for 10 min. Then, EDC.HCl and NHS (20 equivalents each) were added to the reaction mixture and stirred for 60 min at 2–8 °C. A 7.4 pH adjustment was made to the reaction mixture. Finally, SDSSD, previously dissolved in Milli-Q water, was added to the reaction mixture and stirred overnight. The resultant mixture was dialyzed using a 2KD dialysis membrane for 48 h to remove unconjugated SDSSD. Purified SDSSD-DSPE (conjugate) was freeze-dried and characterized by ^1^H-NMR, Maldi-TOF, and a Scanning Electron Microscope (SEM) followed by Energy Dispersive X-ray Analysis (EDX). Figure 1 represents the conjugation scheme, whereas Appendix A depicts the conjugation procedure for SDSSD-DSPE.

#### Determination of % Conjugation Efficiency

In order to determine the % conjugation efficiency, a reaction mixture containing SDSSD-DSPE and unconjugated SDSSD was injected directly into HPLC (HPLC 1260 Infinity, Agilent Technologies, Santa Clara, CA, USA). Eclipse Plus C18 column (100 Å, 4.6 mm × 250 mm, 5 µm) was utilized for separation using a mobile phase consisting of 0.1 % *v*/*v* formic acid in acetonitrile and 0.1 % *v*/*v* formic acid in Milli-Q as an organic phase and aqueous phase respectively. The analysis was done at 210 nm by keeping the injection volume at 10 μL. Appendix A represents the standard chromatogram of SDSSD, whereas Appendix A depicts the linearity for SDSSD to determine unconjugated SDSSD concentration, thereby determining % conjugation efficiency.

### 2.3. Analytical Method for PTH (1-34)

PTH (1-34) was quantified using a validated HPLC method during the formulation development process [33]. PTH (1-34) from SDSSD-LPs was determined by using HPLC (HPLC 1260 Infinity, Agilent Technologies, Santa Clara, CA, USA). Chromatographic separation was achieved using XBridge BEH C18 column (300 Å, 4.6 mm × 150 mm, 10 μm) and a mobile phase consisting of 0.1 % *v/v* formic acid in acetonitrile and 0.1 % *v*/*v* formic acid in Milli-Q as an organic phase and aqueous phase, respectively. The analysis was carried out at 210 nm with an injection volume of 50 μL.

### 2.4. Development of SDSSD-LPs

SDSSD-LPs were fabricated using the ethanol injection method as per our previous method with a slight modification [34]. Briefly, the organic phase was made by dissolving HSPC, cholesterol, DSPE-PEG2000, and SDSSD-DSPE in absolute ethanol and injected into 3 mL of acetate buffer containing PTH (1-34) with continuous stirring on a magnetic stirrer at 500 RPM. Nanoprecipitation occurs when the organic phase comes into contact with the drug-containing aqueous phase, resulting in the formation of PTH (1-34) encapsulated nanoliposomes (SDSSD-LPs). Figure 2 represents the schematic diagram of the preparation of SDSSD-LPs. The injection rate was kept at 1 mL per minute. After 15 min of stirring, the developed dispersion of SDSSD-LPs was centrifuged at 50,000 RPM for 60 min to separate unencapsulated PTH (1-34). The obtained pellet of SDSSD-LPs was redispersed in water and kept at −20 °C until use. Unconjugated liposomes (LPs) were formulated similarly to the above-mentioned method without the addition of SDSSD-DSPE. For confocal studies, fluorescent SDSSD-LPs were prepared by adding 0.5 mM NBD-PE in the organic phase.

### 2.5. Optimization of SDSSD-LPs

For the optimization of SDSSD-LPs, a three-factor Central Composite Design (CCD) was explored to assess the impact of independent factors/variables on two dependent factors/variables. The selection of independent variables, such as material attributes and process parameters, for the statistical optimization of SDSSD-LPs was carried out using a literature review and prior experimentation. Based on our previous experiment of screening design [35], optimization of non-pegylated [36] and pegylated nanoliposomes [34], drug concentration, lipid concentration, and cholesterol concentration were chosen as independent factors, while the dependent variables for the optimization of SDSSD-LPs were particle size and % entrapment efficiency (% EE) by following the response surface method (RSM).

Selected material attributes and process parameters with their ranges are mentioned in Table 1. For generating the experimental design and data analysis, Design-Expert software (Version 12, Stat-Ease, Inc., Minneapolis, MN, USA) was used. Appendix A represents the layout of the experimental design for the optimization of SDSSD-LPs. The data were analyzed using the response surface regression method. The selection of the polynomial model was carried out using Design Expert software’s significant terms (*p* < 0.05), coefficient of variance, multiple correlation coefficients, and least significant lack of fit.

#### 2.5.1. Statistical Analysis for CCD

The optimum concentration of the independent factors (drug, lipid, and cholesterol) for SDSSD-LPs formulation was selected depending upon the requirement of minimum particle size and maximum % EE. The behavior of the response surface was evaluated for the selected response function (dependent variable) using a polynomial equation. The equation below describes the generalized response surface model, wherein y stands for the predicted response; *β*_0_ is constant, and *β*_1_ and *β*_2_ are the linear, quadratic, and interaction coefficients, respectively [37].
*Y_i_* = *β*_0_ + *β*_1_*x*_1_ + *β*_2_*x*_2_ + *β*_11_*x*_1_^2^ + *β*_22_*x*_2_^2^ + *β*_12_*x*_1_*x*_2_(1)

The significant difference between independent variables was obtained using analysis of variance (ANOVA). The model reduction was evaluated on the basis of the model *p*-value and Box-Cox plot for the power transform. The effects of all significant independent variables (*p* < 0.05) were taken into account in the reduced model. Additionally, a residual analysis was conducted to examine the behavior of residuals. All factor plots, three-dimensional response surface plots, and contour plots were examined to visualize the influence of the interactions of the variables on the responses. Further, the overlay plot (design space) was composed to get the optimized composition for SDSSD-LPs.

#### 2.5.2. Model Verification

Following the experiment and data evaluation, the obtained predicted response values were compared with the experimental value, and the following calculation was used to get the % residual value:% Residual = (Predicted results − Observed results)/(Predicted results) × 100(2)

### 2.6. Characterization of SDSSD-LPs

#### 2.6.1. Particle Size and Zeta Potential

Particle size and zeta potential of SDSSD-LPs were measured using the Malvern Zetasizer Nano ZS 90 (Malvern Instrument, Malvern, UK). In order to achieve proper scattering intensity, the dispersion of optimized SDSSD-LPs was diluted 100 times in Milli-Q water. Measurements were carried out in a polystyrene cuvette with a 90° scattering angle at 25 °C. Undiluted dispersion of SDSSD-LPs was used to determine zeta potential at 25 °C. All the measurements were recorded in triplicate.

#### 2.6.2. Determination of % Entrapment Efficiency

The % EE of SDSSD-LPs was determined using HPLC as per Section 2.3. The indirect method was used for the estimation of encapsulated PTH (1-34) in the SDSSD-LPs. In detail, the dispersion of SDSSD-LPs was centrifuged at 50,000 RPM for 60 min. The obtained supernatant was gathered and injected into HPLC. The unknown concentration of PTH (1-34) in the supernatant was calculated from linearity, and % EE was determined using the following equation.
% EE = (Amount of total drug − Amount of unentrapped drug)/(Amount of total drug added) × 100(3)

#### 2.6.3. Morphological Assessment

Morphological assessment of SDSSD-LPs was performed using cryo FE-SEM (SIGMA S300, Zeiss, Jena, Germany). For the analysis, the dispersion of SDSSD-LPs was transferred onto rivets mounted on a cryo-SEM sample holder. Freezing of samples was carried out by submerging them into liquid nitrogen. The frozen samples were collected and fractured with the help of a cold knife to remove the extra sample. Sublimation of fractured samples was done at −90 °C for 10 min. Images of SDSS-LPs were acquired following the samples’ exposure to the cryo FE-SEM chamber.

#### 2.6.4. In Vitro Drug Release

In vitro PTH (1-34) release from SDSSD-LPs was determined by a separate sample method. Briefly, optimized SDSSD-LPs pellets were redispersed in PBS (pH 7.4). 1 mL of the prepared dispersion was added into a microcentrifuge tube. A single microcentrifuge tube was considered a single time point for the drug release study. All tubes were incubated at 37 °C at 100 RPM in an orbital shaker. At each time point, the microcentrifuge was withdrawn from the shaker and centrifuged to separate the released drug. The released drug concentration was determined by HPLC. All instrumental parameters were kept the same as per Section 2.3 for the quantification of PTH (1-34).

#### 2.6.5. In Vitro Bone Mineral Binding Assay

To determine the bone binding potential of optimized SDSSD-LPs, an in vitro bone mineral binding assay was performed as per the previous report with a slight modification [38]. First, PTH (1-34) encapsulated SDSSD-LPs and LPs (unconjugated formulation) were prepared. The HA crystals were dispersed in PBS (pH 7.4), and a 300 μL dispersion of NBD-PE loaded SDSSD-LPs, and LPs was added into an HA-containing tube separately. Both samples were mixed properly and incubated at 37 °C for 60 min. Then, the samples were centrifuged at 10,000 RPM for 5 min. Fluorescence measurement of the supernatant was carried out by a multimode UV microplate reader (Varioskan LUX, Thermo Fischer Scientific, Waltham, MA, USA) using excitation and emission wavelengths of 463 nm and 536 nm, respectively. Figure 3 demonstrates the procedure for an in vitro bone mineral binding assay.

#### Confocal Analysis

The supernatant from both samples was discarded, and pellets from both samples were collected. The drying of these samples was carried out in the dark at room temperature, as NBD-PE is highly light-sensitive. Fluorescent images of dried hydroxyapatite crystals were captured by a laser-scanning confocal microscope (LSCM) (Leica, Wetzlar, Germany) using excitation and emission wavelengths of 463 nm and 536 nm, respectively. In order to perform a semi-quantitative analysis, the images were processed using Image J software LSCM and Corrected Total Cell Fluorescence (CTCF) was derived for comparing the bone mineral binding potential of SDSSD-LPs and LPs.

### 2.7. Statistical Analysis

GraphPad Prism was used for the statistical analysis of the results (version 6.0, Foster City, CA, USA). All outcomes are displayed as mean ± standard deviation. (n ≥ 3). The student *t*-test was used to establish statistical significance. The following definitions apply to the derived *p*-values: * *p* < 0.05, ** *p* < 0.01, *** *p* < 0.001, **** *p* < 0.0001. The letters n.s. stand for non-significant differences.

## 3. Results and Discussion

### 3.1. Synthesis and Characterization of SDSSD

The synthesis of SDSSD was carried out using solid-phase peptide synthesis using standard Fmoc chemistry. Figure 4 represents the chemical structure of SDSSD. Amino acids were coupled on chlrotrityl resin and subsequently cleaved by TFA-Water. The semi-preparative HPLC method was used to purify the crude peptide and was characterized by mass spectrometry and ^1^H NMR. Appendix A depicts the mass spectra of SDSSD obtained from a mass spectrometer in positive mode. The experimental mass value of 510.17 [M+H]^+^ was found to be the same as the theoretical value (509.16). Therefore, it confirmed the successive synthesis of SDSSD. Additionally, confirmation of SDSSD was also done by ^1^H NMR. Appendix A shows the characteristic peak for SDSSD, which appeared at 8 PPM (corresponding to four -NH) and the peak close to 5 PPM (corresponding to -NH_2_). This was further confirmed by the hydrogen-deuterium exchange study (Appendix A).

### 3.2. Characterization of SDSSD-DSPE

The conjugation between SDSSD and DSPE-PEG2000-COOH was performed in the presence of EDC and NHS in order to obtain an acyl amino ester that could further react with the primary amine group of SDSSD and yield an amide bond between SDSSD and the linker, DSPE-PEG2000-COOH. Further, the developed conjugate was characterized by NMR, Maldi-TOF, and SEM/EDX.

#### 3.2.1. ^1^H NMR

The conjugation of SDSSD-DSPE was confirmed by ^1^H NMR. The peaks appeared at 7.8 PPM for SDSDD-DSPE, with six proton integration resulting from four amide protons (at 8 PPM) from SDSSD (Appendix A), one amide from DSPE-PEG2000-COOH (8 PPM) (Appendix A), and one proton from the newly formed amide bond between SDSSD and DSPE-PEG2000-COOH (Figure 5). Hence, ^1^H NMR demonstrated the successive synthesis of SDSSD-DSPE.

#### 3.2.2. Maldi TOF

The SDSSD-DSPE was also confirmed by Maldi-TOF. The molecular weights of DSPE-PEG2000-COOH and SDSSD were 2778.69 and 509.16, respectively. In the conjugation process, one water molecule was removed to form the amide bond between SDSSD and DSPE-PEG2000-COOH. The theoretical mass of SDSSD-DSPE is 3247.86. Figure 6 shows that the molecular weight of the obtained SDSSD-DSPE is 3430.71, which means the distribution had its mode at 3430.71. The difference in molecular weight of SDSSD-DSPE between the theoretical mass and the experimental value obtained from Maldi-TOF is more likely due to the polydispersity of PEG2000 [39]. Additionally, the manufacturer’s certificate of analysis makes reference to it. The bell-shaped distribution of the conjugate’s molecular mass was validated by this mass spectrometry. These results confirm the successive conjugation of SDSSD with DSPE-PEG2000-COOH to obtain SDSSD-DSPE.

#### 3.2.3. SEM/EDX

SDSSD-DSPE was further confirmed by SEM/EDX analysis. The elemental composition of freeze-dried SDSSD-DSPE was compared with DSPE-PEG2000-COOH. The conjugation was confirmed based on the elemental composition. The N content of DSPE-PEG2000-COOH is mainly derived from one amide group, whereas the N content of SDSSD-DSPE is derived from one amide of DSPE-PEG2000-COOH, four amide groups of SDSSD, and a one newly formed amide linkage between SDSSD and DSPE-PEG2000-COOH. Figure 7 demonstrates the elemental analysis using SEM/EDX. Table 2 represents the elemental composition of DSPE-PEG2000-COOH and SDSSD-DSPE. During the sample preparation, both samples of approximately equal weight were attached to carbon tape for analysis. Therefore, during data analysis, carbon area and % carbon content were excluded from the final results. The SEM/EDX analysis demonstrated the elemental composition of SDSSD-DSPE and further confirmed the synthesis of SDSSD-DSPE.

#### 3.2.4. Percentage Conjugation Efficiency of SDSSD-DSPE

The percentage conjugation efficiency of SDSSD-DSPE was determined by the HPLC. Unconjugated SDSSD content was measured from the reaction mixture, and % conjugation was derived from reducing the obtained SDSSD from the initially added content. The % conjugation of SDSSD-DSPE was found to be 47.36 ± 1.09%. The obtained results indicated that the EDC/NHS reaction achieved around 50 % SDSSD conjugation with DSPE-PEG2000-COOH, which is sufficient to target the PTH (1-34) to the bone. Appendix A represents the chromatogram used for the determination of the % conjugation efficiency.

#### 3.2.5. Optimization of SDSSD-LPs

Optimization of SDSSD-LPs was carried out by CDD, a statistical method. Independent variables such as drug concentration, lipid concentration, and cholesterol concentration were selected based on the literature review and preliminary experiments. Stirring speed, DSPE-PEG2000, and SDSSD-DSPE concentrations were kept constant as their concentrations were very low in the formulation.

In all trials, the stirring speed was kept at 500 RPM. In the ethanol injection method, the formation of lipidic vesicles occurs spontaneously. Two different independent variables (response) were recorded (particle size and % EE). In order to assess the impact of independent factors on response, CCD was composed using Design-Expert software (Version 12, Stat-Ease, Inc., Minneapolis, MN, USA). The layout of the design and experimental runs are listed in Appendix A. Several modules were explored to fit the experimental data. For the purpose of estimating the best-fit model for the independent variables of SDSSD-LPs, all the data were statistically examined. ANOVA was utilized to determine the significance of the developed models. The higher F-value and minimum *p*-value of terms in the model suggested a significant effect on dependent variables.

##### Effect of Independent Variables on the Particle Size

Particle size is a critical characteristic of the liposomal product. Model F-value and *p*-value were found to be 6.18 and 0.01, respectively, which indicates the model and model terms are significant. An indication that the model terms are not significant is when the *p*-value is higher than 0.10. Generally, model reduction is required when there are several insignificant model terms. Based on the *p*-value of the model and the lack of fit value (0.79), it was found that independent variables influence particle size significantly. The Lack of Fit F-value of 0.55 suggests that the Lack of Fit is not significant in comparison to the pure error. Further, the adjusted R^2^ of 0.39 and the predicted R^2^ of 0.22 are reasonably congruent. Therefore, the difference is less than 0.2, which was confirmed by fit statistics. The adequate signal was found to be 7.40, which measures the signal-to-noise ratio. This suggests that this model could aid in navigating the design space. Before the analysis of particle size data from SDSSD-LPs, Residual plots and Box-Cox plots were analyzed. All residuals behaved very well. Figure 8A depicts the Normal plot of Residuals for particle size. The Box-Cox plot was used to check the power transformation (Figure 8B). The statistical information for the model and model terms is compiled in Table 3. In the present study, the ethanol injection method was explored for the preparation of SDSSD-LPs. In this method, when the organic phase containing lipids comes in contact with an aqueous solution, it results in the spontaneous formation of nanoliposomes [40].

The particle size of SDSSD-LPs was obtained between 52.55 to 280.20 nm. It has been stated that increasing lipid concentration leads to a rise in the particle size of vesicles [40]. Appendix A represents the contour plot and 3D response plot for the particle size of SDSSD-LPs. Appendix A and Figure 9B demonstrate that as lipid concentration increases, the particle size of SDSSD-LPs also increases. This larger particle size may be observed due to the increased viscosity of the organic phase. A higher viscous solution hinders the diffusion of organic solution into a drug containing an aqueous phase, resulting in large vesicle formation [41]. Cholesterol concentration has a minimal effect on the particle size of SDSSD-LPs (Figure 9C), whereas drug concentration showed a curvature effect (Appendix A and Figure 9A).

##### Effect of Independent Variables on %EE

The percentage of EE of SDSSD-LPs was found to be between 30.18 to 80.28%. Model F-value and *p*-value were found to be 5.58 and 0.03 (Table 3), respectively, which indicates that the model and model terms are significant. The significant model and model terms were determined using the simplified linear model. The Lack of Fit F-value of 2.87 implies that the Lack of Fit is not significant. Further, the Predicted R² of 0.08 is in reasonable agreement with the adjusted R^2^ of 0.22. Table 3 enumerates all the statistical terms for model selection. The difference between the predicted R^2^ and the adjusted R² is less than 0.2. Additionally, an adequate precision was found to be 6.27, indicating that this model can be used for design-space navigation. Additionally, the need for data transformation was evaluated by the Boc-Cox plot, which suggested no transformation during the data analysis for % EE (Figure 8D).

SDSSD-LPs were formulated by injecting a lipidic solution into the aqueous solution of PTH (1-34). Therefore, lipid concentration has an important role in SDSSD-LPs development. In the present formulation, the % EE was significantly influenced by the lipid content. Increased lipid concertation in the organic phase resulted in an improved % EE of SDSSD-LPs, which is depicted from one factor (Figure 9E), contour plots (Appendix A), and 3D surface plots (Appendix A). The % EE increases as the lipid content of the formulation increases [41]. A high concentration of lipids resulted in the maximum number of vesicles, and therefore, it entraps more amount of the drug. Hence, it improves the % EE. An increase in cholesterol concentration may have an impact on % EE. Therefore, the optimum concentration of cholesterol should be used in liposomal formulations. In our study, cholesterol showed the minimum effect on % EE (Figure 9F and Appendix A). The method of preparation also contributes to % EE. It has been observed that the % EE of the hydrophilic drug is less than that of a hydrophobic drug in the ethanol injection method [42]. A method such as the rotary evaporation method forms multilamellar vesicles, which provide more surface area for the encapsulation of hydrophilic drugs [43].

#### 3.2.6. Model Verification

After the analysis of both responses (particle size and % EE), an overlay plot (design space) was generated to get the optimized composition of SDSSD-LPs (Figure 10). An overlay plot was generated based on the target ranges of responses. The target ranges were put into the Design Expert software to achieve the predicted composition of SDSSD-LPs. The targeted response criteria for SDSSD-LPs were particle size less than 200 nm and a maximum % EE. Three predicted trials were selected randomly from the design space.

Optimized composition for SDSSD-LPs was obtained by using drug concentration (21.22 ± 0.86 µM), lipid concentration (48.82 ± 1.42 mM), cholesterol concentration (19.9 mM), DSPE-PEG2000 (2 mM), SDSSD-DSPE (1 mM), and stirring speed (500 RPM). An optimized composition for SDSSD-LPs was evaluated in triplicate. Table 4 shows the optimized composition for SDSSD-LPs. Experimental response values are expressed as the mean value. The obtained experimental values were matched with the software-predicted values. Further, % residual values were measured to evaluate the predictability of the model. No significant difference was found between expected and observed values. Hence, the developed model served the purpose of the statistical development of SDSSD-LPs.

### 3.3. Characterization of SDSSD-LPs

#### 3.3.1. Particle Size and Zeta Potential

The particle size and zeta potential of SDSSD-LPs were determined. The particle size of optimized SDSSD-LPs was observed to be 185.17 ± 1.4 nm. The single peak of the particle size distribution of SDSSD-LPs represents the even distribution of liposomes in the optimized formulation. The particle size distribution of SDSSD-LPs is depicted in Figure 11A. The zeta potential of SDSSD-LPs was determined to be −25.03 ± 0.21 mV. The zeta potential of the optimized formulation is presented in Figure 11B.

#### 3.3.2. % EE

To determine the concentration of entrapped drug within optimized SDSSD-LPs, % EE was evaluated. The validated HPLC method was used to determine the % EE of the formulation. The optimized SDSSD-LPs had an % EE of 66.72 ± 4.22%.

#### 3.3.3. Morphological Characterization

The optimized SDSSD-LPs were characterized by cryo FE-SEM. These techniques help to understand surface characteristics, including surface morphology and the size of nanoformulation. The images acquired from cryo FE-SEM demonstrated that SDSSD-LPs have a spherical shape and smooth surface. Figure 12 depicts the surface morphology of SDSSD-LPs derived by cryo FE-SEM.

#### 3.3.4. In Vitro Drug Release

In vitro PTH (1-34) release from SDSSD-LPs was carried out using a separate sample method. Figure 13 depicts the release profile of PTH (1-34). In PBS, SDSSD-LPs demonstrated a sustained release profile (pH 7.4). This sustained release of PTH (1-34) from SDSSD-LPs might be due to the lipidic barriers of nanoliposomes. It was found that at 3 h, more than 50% of the PTH (1-34) was released from SDSSD-LPs which reached up to 66% at 6 h and 75% at 12 h, which further reached up to 80% at 24 h.

#### 3.3.5. In Vitro Bone Mineral Binding Assay

The main constituents of the skeleton are HA crystals. It is well-reported that several oligopeptides show a strong affinity for the bone mineral HA. In particular, an aspartic acid peptide (a negatively charged amino acid) containing oligopeptide has a strong affinity for bone tissue. Negatively charged amino acids bind to the calcium to reach the bone surface. Therefore, to determine the binding capacity of optimized SDSSD-LPs to bone tissue, an in vitro bone mineral binding assay (HA binding assay) was carried out using NBD-PE loaded SDSSD-LPs. After a 60-min incubation, the fluorescence of the supernatant was analyzed. Figure 14 shows the % relative fluorescence and images for SDSSD-LPs and unconjugated LPs binding to HA crystals. Figure 14A depicts less fluorescence in the supernatant of SDSSD-LPs incubated with HA crystals resulting from the binding of SDSSD-LPs, whereas unconjugated formulations (LPs) showed higher fluorescence in supernatants. After discarding the supernatant, pellets of HA crystals also showed a more colorful appearance in SDSSD-LPs compared to the unconjugated formulation (Figure 14B).

Further, confocal image analysis also showed SDSSD-LPs bound HA crystals have more fluorescence compared to the unconjugated LPs (Figure 15A,B). Additionally, semi-quantitative analysis was carried out using ImageJ software, which also suggested that HA crystals incubated with SDSSD-LPs have more fluorescent intensity compared to the nonconjugated formulation (Figure 15C). Therefore, these results indicated that the SDSSD-LPs could precisely target the PTH (1-34) on bone surfaces.

## 4. Conclusions and Future Perspective

SDSSD peptide conjugated nanoliposomes were fabricated for bone-targeted delivery of PTH (1-34). To achieve this, in-house SDSSD was synthesized and conjugated with the lipidic linker. The developed conjugate was characterized using various analytical methods to confirm the conjugation. SDSSD-LPs were prepared using the ethanol injection method and optimized by using a CCD statistical design. Further, the effects of various formulation and process parameters/variables on dependent variables were analyzed. The particle size and zeta potential of the optimized formulation were observed to be 185.17 ± 1.4 nm and −25.03 ± 0.21 mV, respectively. Morphological analysis revealed that developed nanovesicles were uniform. SDSSD-LPs exhibited a sustained release profile of PTH (1-34). Moreover, the bone mineral binding assay showed that SDSSD-LPs have a higher binding potential to HA crystals compared to nanoliposomes without SDSSD modification. Further, to evaluate the in vivo safety and efficacy of the developed formulation, extensive in vivo studies need to be carried out. After the successful post-pre-clinical evaluation, the developed novel carrier system can be assessed in a clinical setting to establish the potential of SDSSD-LPs in the treatment of osteoporotic conditions. Thus, SDSSD-LPs are expected to become a potential approach for bone-targeted drug delivery in osteoporotic treatment.

## Figures and Tables

**Figure 1 pharmaceutics-15-00608-f001:**
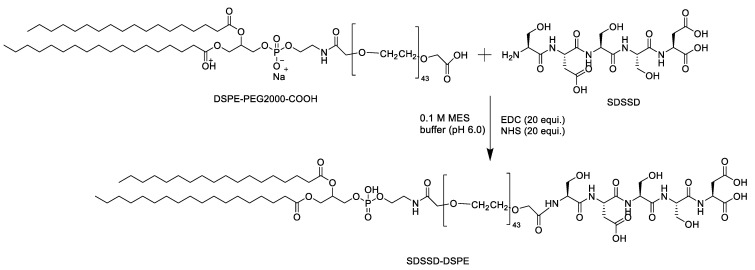
Synthesis scheme for the preparation of SDSSD-DSPE.

**Figure 2 pharmaceutics-15-00608-f002:**
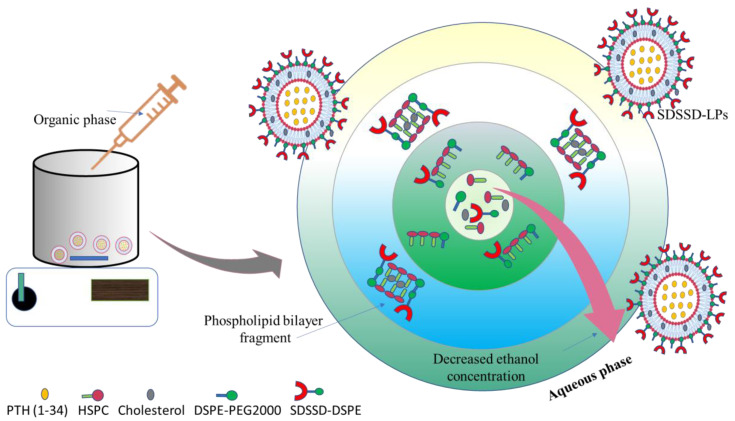
Schematic representation for preparation of SDSSD-LPs. The lipidic solution containing HSPC, cholesterol, DSPE-PEG2000, and SDSSD-DSPE, when injected into PTH (1-34) containing aqueous solution, diffusion of organic phase take place which resulted in nanoprecipitation of lipids at organic- aqueous interface thereby formation of nanoliposomes.

**Figure 3 pharmaceutics-15-00608-f003:**
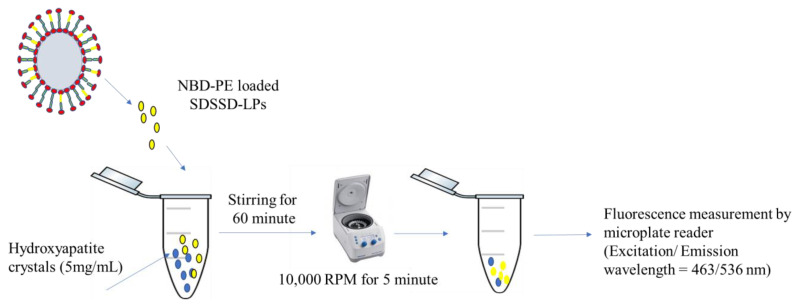
Schematic representation of in vitro bone mineral binding assay.

**Figure 4 pharmaceutics-15-00608-f004:**
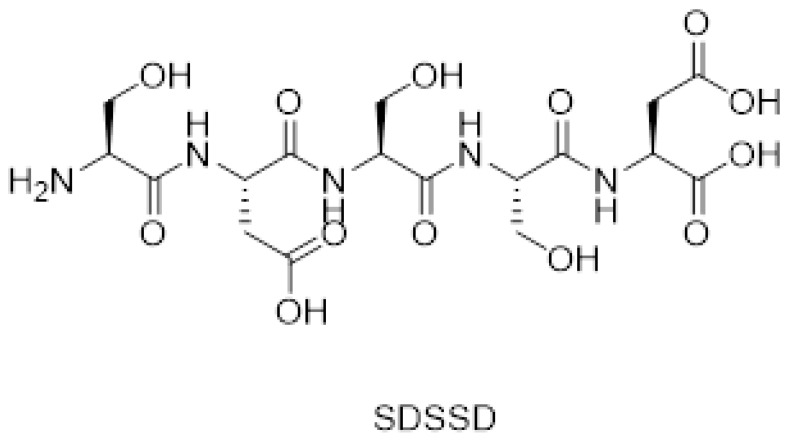
Chemical structure of SDSSD.

**Figure 5 pharmaceutics-15-00608-f005:**
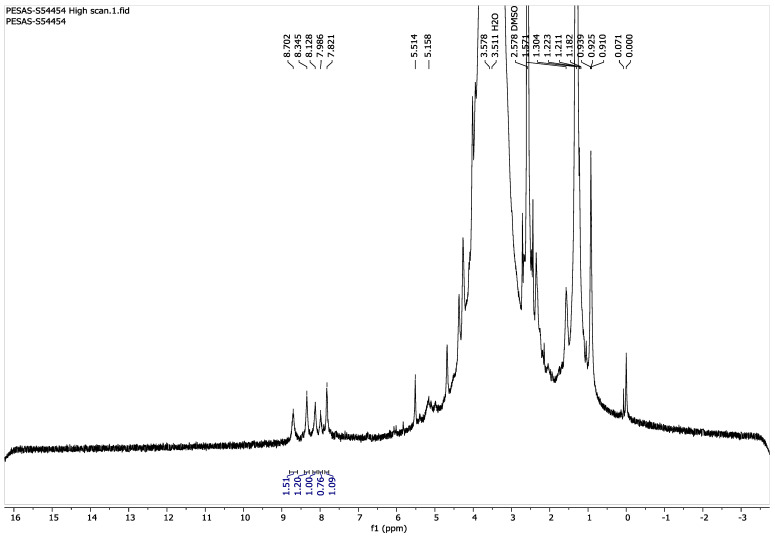
^1^H NMR of SDSSD-DSPE.

**Figure 6 pharmaceutics-15-00608-f006:**
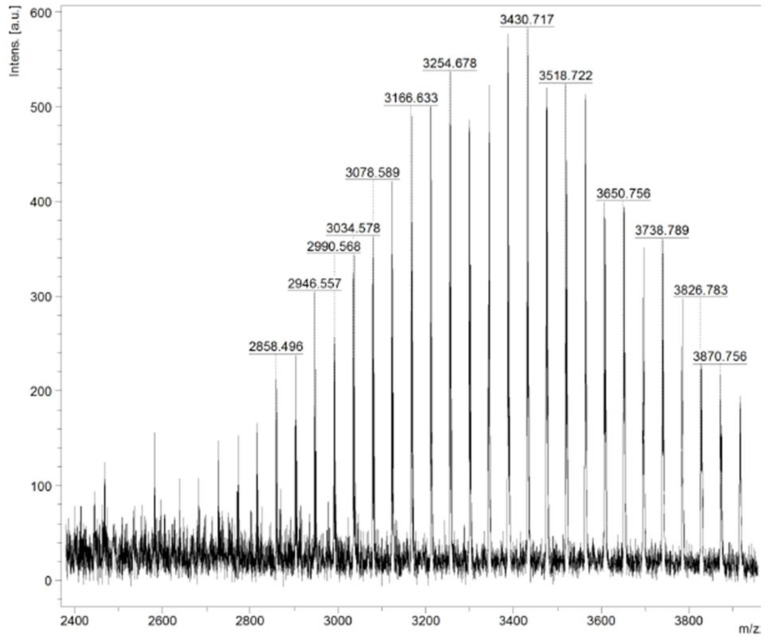
Maldi-TOF analysis of SDSSD-DSPE.

**Figure 7 pharmaceutics-15-00608-f007:**
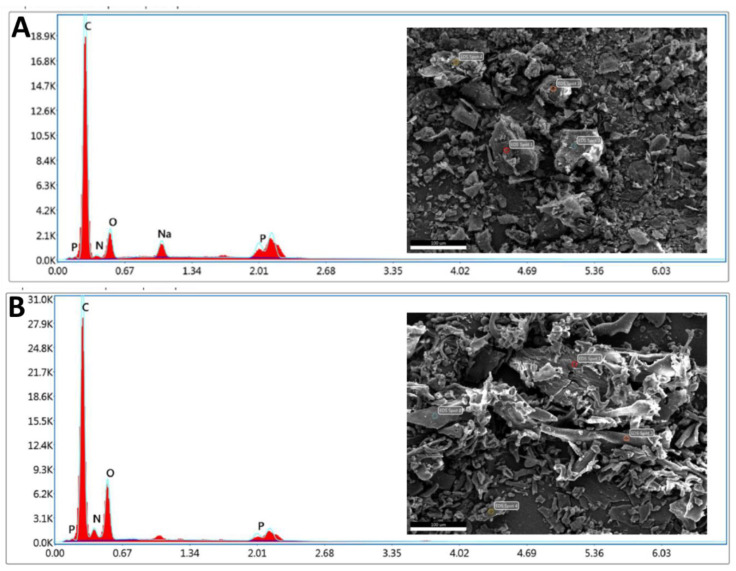
Elemental analysis by SEM/EDX. (**A**) DSPE-PEG2000-COOH, (**B**) SDSSD-DSPE. Scale bar: 100 μm.

**Figure 8 pharmaceutics-15-00608-f008:**
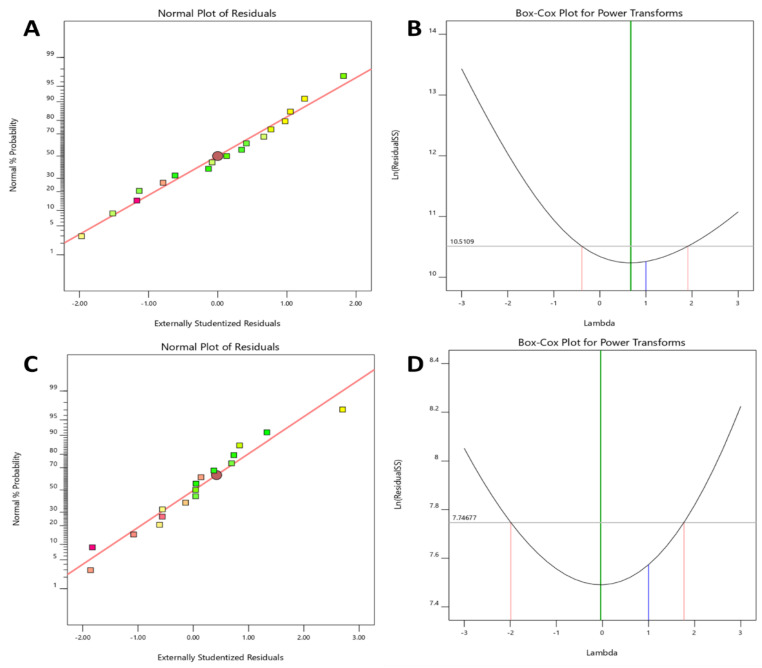
Residual analysis and power transformation plots. (**A**) The normal plot of residual for particle size, (**B**) Boc-Cox plot for power transformation for particle size. (**C**) The normal plot of residual for % EE, and (**D**) Boc-Cox plot of power transformation for % EE.

**Figure 9 pharmaceutics-15-00608-f009:**
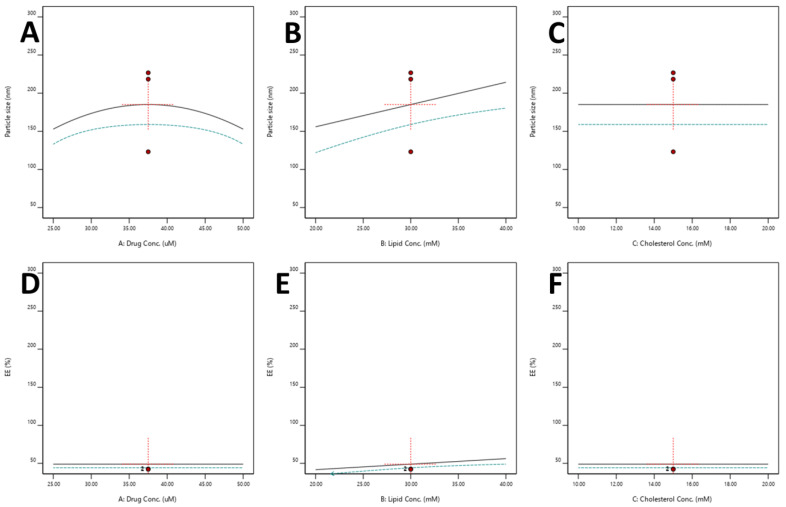
All factor graph; (**A**–**C**) All factor plots for particle size, (**D**–**F**) All factor plots for % EE.

**Figure 10 pharmaceutics-15-00608-f010:**
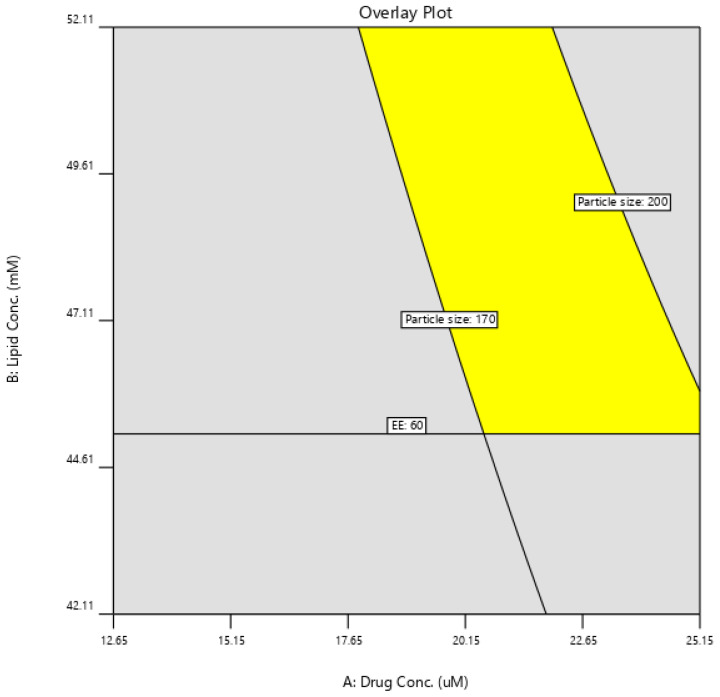
Design space for SDSSD-LPs.

**Figure 11 pharmaceutics-15-00608-f011:**
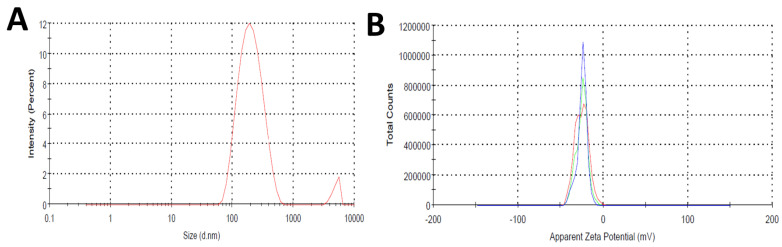
Particle size (**A**,**B**) zeta potential of SDSSD-LPs. The data represent the means ± SD (n = 3).

**Figure 12 pharmaceutics-15-00608-f012:**
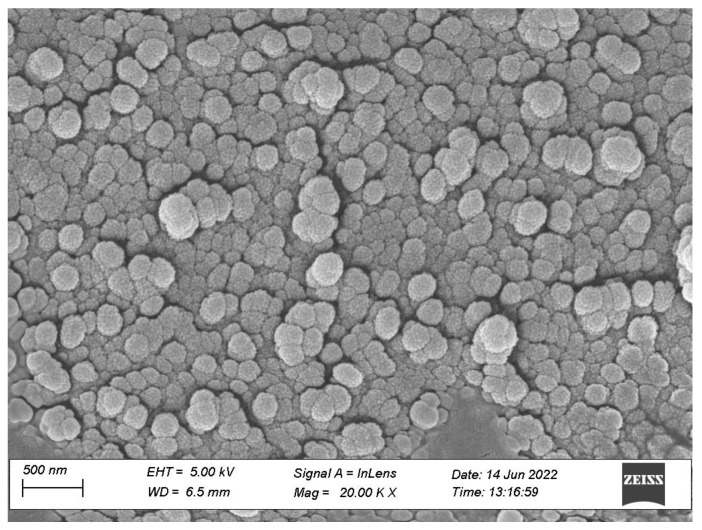
Morphological assessment of SDSSD-LPs by cryo-SEM. Scale bar: 500 nm.

**Figure 13 pharmaceutics-15-00608-f013:**
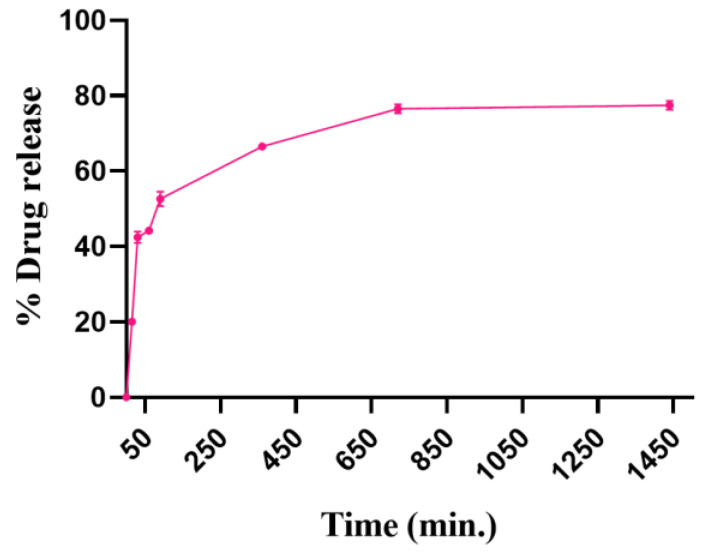
The in vitro release profile of PTH (1-34) from SDSSD-LPs. Data represented the mean ± SD (n = 3).

**Figure 14 pharmaceutics-15-00608-f014:**
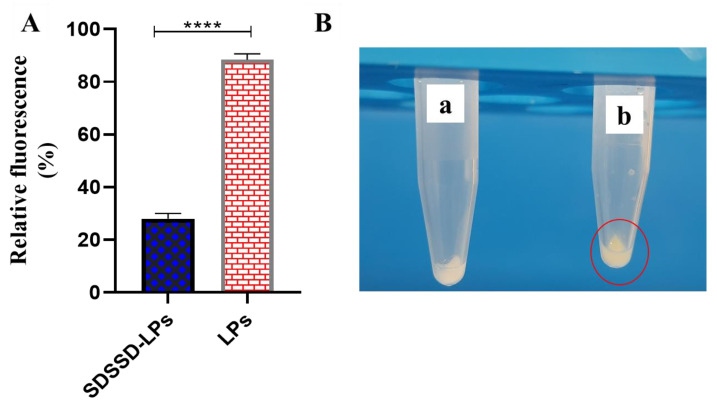
In vitro bone mineral binding assay. (**A**) % relative fluorescence of supernatant; (**B**) Comparative image of binding of un-conjugated LPs (**a**) and SDSSD-LPs (**b**) to HA crystals. The findings illustrate the means ± SD (n = 6). **** *p* < 0.0001.

**Figure 15 pharmaceutics-15-00608-f015:**
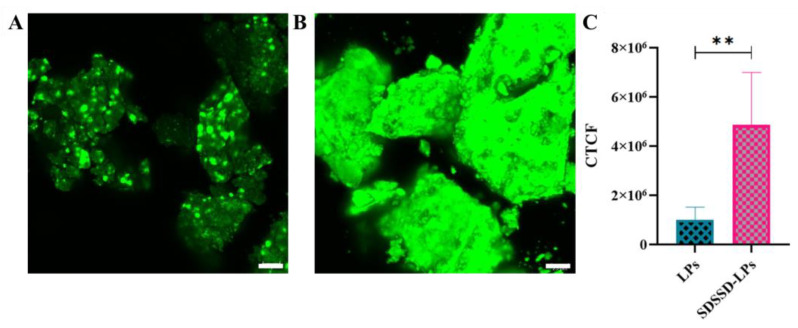
Confocal analysis. (**A**) LSCM image of unconjugated LPs, (**B**) LSCM image of SDSSD-LPs, and (**C**) Semi-quantitative analysis using Image J. The results represent the means ± SD (n = 3). ** *p* < 0.01. Scale bar: 25 μm.

**Table 1 pharmaceutics-15-00608-t001:** Independent variables with high and low levels for SDSSD-LPs.

Factor	Name	Units	Levels	
			Low (−1)	High (+1)
1	Drug concentration	μM	16.48	58.52
2	Lipid concentration	mM	13.18	46.82
3	Cholesterol concentration	mM	6.59	23.41
Constant parameters				
4	DSPE-PEG2000	mM	2	
5	SDSSD-DSPE	mM	1	
6	Stirring rate	RPM	500	

**Table 2 pharmaceutics-15-00608-t002:** Elemental analysis by SEM/EDX.

Compound	DSPE-PEG2000-COOH		SDSSD-DSPE	
Element counts	Nitrogen counts (NK)	Oxygen counts (OK)	Nitrogen counts (NK)	Oxygen counts (OK)
Weight %	2.68	12.56	7.80	25.89
Atomic %	2.58	10.58	7.42	21.58

**Table 3 pharmaceutics-15-00608-t003:** Statistical data of model terms.

Terms	Particle Size (nm)	EE (%)	Inference
Model *p*-value	0.01	0.03	Significant
Model F value	6.18	5.58	
Lack of Fit	0.79	0.29	Non-significant
R^2^	0.47	0.27	
Adjusted R^2^	0.39	0.22	
Predicted R^2^	0.22	0.08	
Adeq Precision	7.40	6.27	

**Table 4 pharmaceutics-15-00608-t004:** Optimized formulation parameters.

Independent Variables	Values	Responses	Predicted Value	Experimental Values	Residual Values (%)
Drug conc. (µM)	21.22 ± 0.86	Particle size (nm)	185.17 ± 1.43	183.07 ± 0.85	1.13
Lipid conc. (mM)	48.82 ± 1.42	EE (%)	62.64 ± 0.89	66.72 ± 4.22	−6.49

## Data Availability

Not applicable.

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
