# Peer review of "Peptide Engraftment on PEGylated Nanoliposomes for Bone Specific Delivery of PTH (1-34) in Osteoporosis"

_pharmaceutics, 2023, doi:10.3390/pharmaceutics15020608_

Round 1
Reviewer 1 Report
Peptide Engraftment on PEGylated Nanoliposomes for Bone 2Specific Delivery of PTH (1-34) in Osteoporosis
In this paper, the authors claim to have produced a lipid nanoparticle (LNP) formulation containing a bone targeting peptide SDSSD. They further claim that SDSSD conjugated LNP shows enhanced uptake into bone in vitro.
The successful conjugation of SDSSD to the lipid is demonstrated by NMR and Maldi-TOF data presented in Figures 5 and 6. DLS and zeta potential reported in Figure 11 along with the SEM data presented in Figure 12 support the claim that SDSSD-lipid conjugates form lipid nanoparticles. Figure 13 demonstrates the release of PTH from LNP for 12 h. Figure 14 and 15 demonstrate the ability of the SDSSD-LNP to bind hydroxyapatite. Overall, the data support the claims made by the authors.
Minor comments:
1. Figure captions can be more descriptive. For example, figure 11B has multiple traces in different colors. There is no legend or other explanation on how these traces are different from each other.
2. Some statements need rewording. For example,
However, the frequent and repeated injections sternly compromise patient compatibility. Moreover, even after subcutaneous administration, owing to a very short half-life of 1 hour, most of the time in a 24-hour duration remains drug-free.
Author Response
Reviewer 1.
In this paper, the authors claim to have produced a lipid nanoparticle (LNP) formulation containing a bone targeting peptide SDSSD. They further claim that SDSSD conjugated LNP shows enhanced uptake into bone in vitro.
The successful conjugation of SDSSD to the lipid is demonstrated by NMR and Maldi-TOF data presented in Figures 5 and 6. DLS and zeta potential reported in Figure 11 along with the SEM data presented in Figure 12 support the claim that SDSSD-lipid conjugates form lipid nanoparticles. Figure 13 demonstrates the release of PTH from LNP for 12 h. Figure 14 and 15 demonstrate the ability of the SDSSD-LNP to bind hydroxyapatite. Overall, the data support the claims made by the authors.
Minor comments:
- Figure captions can be more descriptive. For example, figure 11B has multiple traces in different colours. There is no legend or other explanation on how these traces are different from each other.
Ans: Three different colours indicated the three separate batches of nanoliposomes for the determination of zeta potential (n=3).
- 2. Some statements need rewording. For example, However, the frequent and repeated injections sternly compromise patient compatibility. Moreover, even after subcutaneous administration, owing to a very short half-life of 1 hour, most of the time in a 24-hour duration remains drug-free.
Ans: As per the reviewer’s suggestion, mentioned statement has been modified and highlighted in revised manuscript.
Reviewer 2 Report
This paper titled “Peptide Engraftment on PEGylated Nanoliposomes for Bone Specific Delivery of PTH (1-34) in Osteoporosis” is a well-written study which deals with the data needed to find a way to deliver drugs in osteoporotic conditions.
However, it appears that some revisions are required for publication in the journal “Pharmaceutics”.
1. English proofreading is required throughout the paper.
- First of all, there are too many typos to count.
In particular, “SDSD-DSPE”, the second word in line 145 of the Material and Method part and “DSPE-PEG000-COOH”, the first word of line 341 of the Result part, were not accurately described even though they are the keywords of this study. Repetition of typos in such an important part can reduce the overall reliability of the paper, so it is necessary to check and correct it as a whole.
- It is necessary to modify the unit notation.
The fourth word “RPM” in line 176 of the Material and Method section is capitalized here, but in line 264 it is written in lower case “rpm”. Also, when writing the measurements, there are many missing parts such as: a space is required between the number and the unit. This part needs correction.
2. Modifying data
- Images of (Figure 10) and (Figure 12) should be replaced with an image with a higher resolution.
- In addition, the scale bars of (Figure 7), (Figure 12) and (Figure 15), are too small and fuzzy to be properly recognized. This part needs to be expressed more clearly.
- The data labels in (Figure 9) are cluttered. It is recommended to capitalize the labels of all six parts, from A to F.
- It is recommended that the time axis of (Figure 13) be changed to "min" to draw a graph, rather than expressed as a decimal point by marking it in "hour".
- The number of digits below the decimal point in (Table 3) are not the same. It is necessary to indicate the same number of digits below the decimal point.
3. Absence of supporting data
Did you submit supporting data?
The data could not be verified when accessing the link below provided to view the supporting data mentioned in the paper.
4. Discussion
The content of the discussion is insufficient. In order to secure expertise, scientifically in-depth discussions need to be additionally described.
Overall, I hope that the quality of the paper and the content of the discussion can be supplemented with above demands so that the paper can be published.
Author Response
This paper titled “Peptide Engraftment on PEGylated Nanoliposomes for Bone Specific Delivery of PTH (1-34) in Osteoporosis” is a well-written study which deals with the data needed to find a way to deliver drugs in osteoporotic conditions. However, it appears that some revisions are required for publication in the journal “Pharmaceutics”.
- English proofreading is required throughout the paper.
Ans: As per the reviewer’s suggestion, whole manuscript has been proofread for language improvement.
- First of all, there are too many typos to count. In particular, “SDSD-DSPE”, the second word in line 145 of the Material and Method part and “DSPE-PEG000-COOH”, the first word of line 341 of the Result part, were not accurately described even though they are the keywords of this study. Repetition of typos in such an important part can reduce the overall reliability of the paper, so it is necessary to check and correct it as a whole.
Ans: As per the reviewer’s suggestion, whole manuscript has been checked for the typos and corrected in revised manuscript. For further clarification; SDSD-DSPE is corrected to SDSSD-DSPE. In Materials and method part, DSPE-PEG2000-COOH and DSPE-PEG2000 are two different components. “DSPE-PEG000-COOH is corrected to “DSPE-PEG2000-COOH. All corrections are highlighted in green color in revised manuscript.
- It is necessary to modify the unit notation.
The fourth word “RPM” in line 176 of the Material and Method section is capitalized here, but in line 264 it is written in lower case “rpm”. Also, when writing the measurements, there are many missing parts such as: a space is required between the number and the unit. This part needs correction.
Ans: As per the reviewer’s suggestion, whole manuscript has been checked for unit notation and corrected in revised manuscript. Further, some missing part suggested by reviewer has also been modified in revised manuscript and highlighted in green colour.
- Modifying data
Images of (Figure 10) and (Figure 12) should be replaced with an image with a higher resolution.
Ans: As per the reviewer’s suggestion, images of (Figure 10) and (Figure 12) has been replaced with an image with a higher resolution in revised manuscript and highlighted in green colour.
- In addition, the scale bars of (Figure 7), (Figure 12) and (Figure 15), are too small and fuzzy to be properly recognized. This part needs to be expressed more clearly.
Ans: As per the reviewer’s suggestion, Figure 7 has been enlarged in revised manuscript, Figure 12 and Figure 15 has been replaced with good resolution with improved scale bars. All changes have been highlighted in green color in revised manuscript.
- The data labels in (Figure 9) are cluttered. It is recommended to capitalize the labels of all six parts, from A to F.
Ans: As per the reviewer’s suggestion, required changes have been made in revised manuscript.
- It is recommended that the time axis of (Figure 13) be changed to "min" to draw a graph, rather than expressed as a decimal point by marking it in "hour".
Ans: As suggested by reviewer, time axis of drug release profile has been changed to min. in revised manuscript.
- The number of digits below the decimal point in (Table 3) are not the same. It is necessary to indicate the same number of digits below the decimal point.
Ans: As per the reviewer’s suggestion, required changes has been made in revised manuscript and highlighted in green colour.
- Absence of supporting data
Did you submit supporting data? The data could not be verified when accessing the link below provided to view the supporting data mentioned in the paper.
Ans: Yes, we had submitted the supporting data but may be due to some technical glitch, data were not accessible. We regret the inconvenience caused to the reviewer for the same, but we would like to mention that it can be accessed.
- Discussion
The content of the discussion is insufficient. In order to secure expertise, scientifically in-depth discussions need to be additionally described.
Ans: As per the reviewer’s suggestion, whole discussion part was verified again and improved in revised manuscript. The newly added or modified discussion part has been highlighted in green colour in revised manuscript.
Overall, I hope that the quality of the paper and the content of the discussion can be supplemented with above demands so that the paper can be published.
Ans: We would like to appreciate reviewer’s valuable comments. All suggestions have been addressed in revised manuscript. We believe that newly added or modified part would full fill the reviewer’s requirement.
Reviewer 3 Report
The loss of BMD leads to bones fragility, moreover we can define osteoporosis when T-score is <-2.5 SD. Strategies based on liposomes seems to be promising in order to delivery drugs in this kind of desease: it reduces off-targets effects of drugs and their toxicity. The main purpose of this article was to design the bone-specific peptide conjugated pegylated nanoliposomes to deliver anabolic drug. In addition, bone-specific-peptide (synthesized by the tecnique of solid phase) is progressively conjugated with a linker (DSPE-PEG2000-COOH).
The authors have presented an interesting topic in the field of peptide functionalized nanocarries. The paper should be considered after minor revisions. It is:
1. Sometime redundant and repetitive in some aspects. There is often reference to the general aspects of the pathology;
2. In this kind of treatment, it is recommended to deepen the synthesis on solid phase;
3. It could be interesting and challenging to study the binding ability of SDSSD-LPs to other stromal structures in order to increase its bone mineral HA’s specificity, such as collagen-based models;
4. Since the in vitro validation was performed on HA crystals, the authors should stress the potential use of these systems in clinical treatment. In this regard a multitude of innovative biocompatible drug delivery systems have been developed, otherwise the systems reported by the authors should provide new avenue in this field. The following references should be added: “bFGF-Loaded Mesoporous Silica Nanoparticles Promote Bone Regeneration Through the Wnt/β-Catenin Signalling Pathway. Int J Nanomedicine. 2022; doi: 10.2147/IJN.S366926”, “Lysyl oxidase engineered lipid nanovesicles for the treatment of triple negative breast cancer. Sci Rep. 2021; doi: 10.1038/s41598-021-84492-3”, “Gold nanoparticles-loaded hydroxyapatite composites guide osteogenic differentiation of human mesenchymal stem cells through Wnt/β-catenin signaling pathway. Int J Nanomedicine. 2019; doi: 10.2147/IJN.S213889”.
Author Response
The loss of BMD leads to bones fragility, moreover we can define osteoporosis when T-score is <-2.5 SD. Strategies based on liposomes seems to be promising in order to delivery drugs in this kind of disease: it reduces off-targets effects of drugs and their toxicity. The main purpose of this article was to design the bone-specific peptide conjugated pegylated nanoliposomes to deliver anabolic drug. In addition, bone-specific-peptide (synthesized by the technique of solid phase) is progressively conjugated with a linker (DSPE-PEG2000-COOH).
The authors have presented an interesting topic in the field of peptide functionalized nanocarriers. The paper should be considered after minor revisions. It is:
- Sometime redundant and repetitive in some aspects. There is often reference to the general aspects of the pathology;
Ans: As per the reviewer’s suggestion, required changes has been made in revised manuscript. Some repetitive sentences have been removed from revised manuscript.
- In this kind of treatment, it is recommended to deepen the synthesis on solid phase;
Ans: As per the reviewer’s recommendation, the synthesis of peptide on solid phase has been described thoroughly in revised manuscript and highlighted in green colour.
- It could be interesting and challenging to study the binding ability of SDSSD-LPs to other stromal structures in order to increase its bone mineral HA’s specificity, such as collagen-based models;
Ans: We appreciate the reviewer’s valuable suggestion. Indeed, it is interesting to study the binding ability of SDSSD-LPs to other stromal structures in order to increase its bone mineral HA’s specificity. Therefore, we are planning to explore these kinds of studies in our future work along with in vivo behaviour of the developed formulation.
- Since the in vitro validation was performed on HA crystals, the authors should stress the potential use of these systems in clinical treatment. In this regard a multitude of I innovative biocompatible drug delivery systems have been developed, otherwise the systems reported by the authors should provide new avenue in this field. The following references should be added: “bFGF-Loaded Mesoporous Silica Nanoparticles Promote Bone Regeneration Through the Wnt/β-Catenin Signalling Pathway. Int J Nanomedicine. 2022; doi: 10.2147/IJN.S366926”, “Lysyl oxidase engineered lipid nanovesicles for the treatment of triple negative breast cancer. Sci Rep. 2021; doi: 10.1038/s41598-021-84492-3”, “Gold nanoparticles-loaded hydroxyapatite composites guide osteogenic differentiation of human mesenchymal stem cells through Wnt/β-catenin signaling pathway. Int J Nanomedicine. 2019; doi: 10.2147/IJN.S213889”.
Ans: As suggested by reviewer, the potential use of developed system has been emphasized in revised manuscript and highlighted in green colour. Further, suggested references also have been included in revised manuscript.
Round 2
Reviewer 3 Report
The paper is now acceptable for the pubblication